# Effect of High-Voltage Electrostatic Field Heating on the Oxidative Stability of Duck Oils Containing Diacylglycerol

**DOI:** 10.3390/foods11091322

**Published:** 2022-04-30

**Authors:** Hailei Sun, Fangfang Li, Yan Li, Liping Guo, Baowei Wang, Ming Huang, He Huang, Jiqing Liu, Congxiang Zhang, Zhansheng Feng, Jingxin Sun

**Affiliations:** 1College of Food Science & Engineering, Shandong Research Center for Meat Food Quality Control, Qingdao Agricultural University, Qingdao 266109, China; sunhailei580231@163.com (H.S.); lff940404@163.com (F.L.); yli@qau.edu.cn (Y.L.); happyglp@126.com (L.G.); wangbw@qau.edu.cn (B.W.); 2National R&D Branch Center for Poultry Meat Processing Technology, Nanjing Huangjiaoshou Food Science and Technology Co., Ltd., Nanjing 211226, China; mhuang@njau.edu.cn; 3Shandong Newhope Liuhe Group Co., Ltd., Qingdao 266000, China; huanghe1@newhope.cn (H.H.); 15965083162@163.com (J.L.); 4Yingyuan Co., Ltd., Jining 272000, China; 13905375260@163.com (C.Z.); f13562431964@163.com (Z.F.); 5Qingdao Special Food Research Institute, Qingdao 266109, China

**Keywords:** high-voltage electrostatic field, duck oil, diacylglycerol, thermal oxidation stability

## Abstract

High-voltage electrostatic field (HVEF) as an emerging green technology is just at the beginning of its use in meat products and by-products processing. In this study, we employed duck oil to produce duck-oil-based diacylglycerol (DAG), termed DDAG. Three different DDAG volume concentrations (0, 20%, and 100%) of hybrid duck oils, named 0%DDAG, 20%DDAG, and 100%DDAG, respectively, were used to investigate their thermal oxidation stability in high-voltage electrostatic field heating and ordinary heating at 180 ± 1 ℃. The results show that the content of saturated fatty acids and trans fatty acids of the three kinds of duck oils increased (*p* < 0.05), while that of polyunsaturated fatty acids decreased (*p* < 0.05) from 0 h to 8 h. After heating for 8 h, the low-field nuclear magnetic resonance showed that the transverse relaxation time (T_21_) of the three oils decreased (*p* < 0.05), while the peak area ratio (S_21_) was increased significantly (*p* < 0.05). The above results indicate that more oxidation products were generated with heating time. The peroxide value, the content of saturated fatty acids, and the S_21_ increased with more DAG in the duck oil, which suggested that the oxidation stability was likely negatively correlated with the DAG content. Moreover, the peroxide value, the content of saturated fatty acids and trans fatty acids, and the S_21_ of the three concentrations of duck oils were higher (*p* < 0.05) under ordinary heating than HVEF heating. It was concluded that HVEF could restrain the speed of the thermal oxidation reaction occurring in the duck oil heating and be applied in heating conditions.

## 1. Introduction

Duck meat is widely favored by consumers due to its special flavor and aroma in China [1] and also for its higher level of unsaturated fatty acids and heme, and higher level of phospholipids and total lipid contents [2,3]. Moreover, the duck by-products of liver, heart, gizzard, and feet are popular in Chinese cuisine. However, duck oil has a relatively poor utilization presently and is most frequently applied as animal feed [4,5].

Oil is an indispensable nutrient in daily diet. However, the consumption of excess oils contributes greatly to obesity and other diseases such as hypertension and cardiovascular diseases [6,7]. It has been reported that duck oil contains a high percentage (over 70.30%) of unsaturated fatty acids [8,9]. Therefore, how to increase the utilization of duck oil for higher economic benefit is becoming a problem for duck abattoirs. We successfully extracted DAG from duck oil in a previous study [5] and intend to create a novel functional oil.

Diacylglycerol (DAG) exists in animal oil and plant oil as a natural ingredient and has an extreme similarity to general oil in flavor [10,11]. Furthermore, DAG could reduce fat buildup in the body; as a novel functional lipid, it is widely used in food processing and is accepted as a safe food by the Food and Drug Administration (FDA) [12,13,14]. DAG has poor oxidation stability due to its rich unsaturated fatty acids and is easily oxidized under ordinary frying conditions [15,16,17]. Hence, it is critical to explore measures to enhance DAG thermal oxidation stability.

High-voltage electrostatic field (HVEF) as a green and safe emerging processing technology has recently drawn a great deal of attention from many food researchers [18,19]. Jia et al. showed that HVEF-assisted freezing significantly enhanced the quality of frozen pork compared with pure freezing [20]. HVEF could restrain the growth of microorganisms in tilapia [21]. Shih et al. reported that HVEF cooking brought a particularly positive effect to the texture and sweet taste of squid [22]. The application of HVEF could resolve the drawbacks of oil refining, such as the loss of bioactive compounds and the elimination of off-flavor, so as to improve the oil-refining process [23]. The HVEF technology has also been used in the extraction of valuable compounds such as proteins, polysaccharides, lipids, etc. [24]. The development of HVEF is still in the early stages, and further study is needed to broaden its applications in the food industry.

Nonetheless, to the best of our knowledge, little information is available on the effect of HVEF on the thermal oxidation stability of oils, especially duck oil and DAG. Therefore, the aim of the present study was to investigate the thermal oxidative stability of different DDAG concentrations (0, 20%, and 100%) of duck oils under high-voltage electrostatic field heating, by testing peroxide value (POV) and color, and using gas chromatography and low-field nuclear magnetic resonance.

## 2. Materials and Methods

### 2.1. Materials

Duck oil was offered by Lichen Oil Co., Ltd. (Weifang, China). Novozym 435 lipase (10,000 U/g) was provided by Novozymes Biotechnology Co., Ltd. (Tianjin, China). 2-thiobarbituric acid (≥98.5%) was obtained by Sinopharm Chemical Reagent Co., Ltd. (Shanghai, China). All other chemical reagents were of analytical or guaranteed purity.

### 2.2. DAG Preparation

DAG extraction was conducted according to our previous study [5]. Briefly, 2000 g duck oil was pre-mixed with isovolumic distilled water and heated for 1.5 h at 87 °C in a water bath (HH-6, Guohua Instrument Manufacturing Co., Ltd., Suzhou, China). After removing water-soluble matters, to the duck oil was added 500 mL NaOH- ethanol solution (1 mol/L), and it was heated for 1 h at 87 °C. Subsequently using NaCl solution to remove glycerol, 10% HCl solution was added to adjust the pH to 2 ~ 3 to release free fatty acids. Thereafter, glycerol (chemical reagent) and free fatty acids were mixed by a volume ratio of 1:2 (glycerol: fatty acids) in a conical flask, 1.65% Novozym 435 lipase was added, and the solution placed into shaker incubator (KYC-100B, Shanghai Fuma Equipment Co., Ltd., Shanghai, China), reacted at 54 °C for 9 h. The final products were separated using a centrifuge (25 °C, 4000 r/min, 20 min) and subsequently the solid sediments were removed. The supernatant was duck-oil-based diacylglycerol (DDAG), in which the concentration of DAG was 86%.

### 2.3. Hybrid Duck Oils Preparation

The hybrid duck oils were prepared by blending the DDAG and duck oil according to different volume ratios (0:1, 1:4, 1:0). The hybrid oils are hereinafter referred to as 0%DDAG, 20% DDAG and 100%DDAG, respectively. The content of DAG in three oils is shown in Table 1.

### 2.4. HVEF Heating

Sample (500 mL) was heated by a high-voltage electrostatic field heater (4 KV, Denba Co., Ltd., Jiaxing, China) and a thermostat-controlled heater (Foshan Nanhai Bofei Mechanical and Electrical Equipment Co., Ltd., Guangdong, China) at 180 ± 1 ℃. Samples were collected every 2 h to be analyzed.

### 2.5. POV

The POV was determined according to the Chinese National Food safety standard method GB 5009.227-2016. In short, 30 mL trichloromethane–glacial acetic acid solution was added to the 2 ~ 3 g sample and the mixture was shaken mildly to totally dissolve the sample; then, to the mixture was added 1 mL standard potash iodide solution, before being shaken for 0.5 min mildly and placed in a dark environment for 3 min. After that, 10 mL of distilled water was added and the 0.01 mol/L Na_2_S_2_O_3_ standard solution was used to titrate. The POV was calculated by the following equation:POV (mmol/kg) = 1000 × (V – V_0_) × c / (2 × m)(1)
where V is the volume of Na_2_S_2_O_3_ standard solution (mL) consumed by the sample, V_0_ is the volume of Na_2_S_2_O_3_ standard solution (mL) consumed in the blank test, c is the concentration of Na_2_S_2_O_3_ standard solution (mol/L), m is the weight of sample (g) and 1000 is the conversion factor.

### 2.6. Color

Color was assessed as described in a previous study, with minor modifications [25]. Color was tested using color difference meter (CR-400, Konica Minolta Holdings Co., Ltd., Tokyo, Japan) for 0, 2, 4, 6, and 8 h. The sample was packed using a transparent sealed bag. Then, the instrument was calibrated with the blackboard and whiteboard and the lightness (*L** value), redness (*a** value) and yellowness (*b** value) of the packed sample were measured.

### 2.7. Gas Chromatography

Gas chromatography was measured according to the Li et al. study with slight modifications [26]. A 60 mg sample was dissolved in 4 mL of isooctane completely. The mixed sample was added to the 200 μL KOH-methanol solution, shaken rigorously for 30 s and placed until it was clear. Then, 1.0 g NaHSO_4_ was added and shaken rigorously. Subsequently, the upper solution after precipitation was filtrated through a 0.22 μm filter membrane. Afterwards, gas chromatography (7890A, Agilent Technologies, Palo Alto, CA, USA), an FID detector, and a CP-Sil 88 capillary column (100 m × 0.25 mm × 0.39 mm, 0.20 μm, Varian Inc, Palo Alto, CA, USA) were used for the analysis. The chromatographic peaks were characterized using the fatty acids methyl ester mixed standard solution. Thereafter, the measurement parameters were set as follows: the injection volume was 1 μL; the injection port temperature was 270 °C; the detector temperature was 280 °C; the split ratio was 100:1; the carrier gas was nitrogen; and the flow mode was constant. The programmed temperature conditions were as follows: 100 °C, 13 min; 100–180 °C, 10 ℃/min, 6 min; 180–200 °C, 1 ℃/min, 20 min; 200–230 °C, 4 ℃/min, 10.5 min.

### 2.8. Low-Field Nuclear Magnetic Resonance (LF-NMR)

The Q-CPMG pulse sequence of a low-field nuclear magnetic resonance analyzer (MicroM.R20-025, Niumai Electronic Technology Co., Ltd., Shanghai, China) was used to determine the transverse relaxation times of the sample. In brief, a 3.0 mL sample was placed in the nuclear magnetic resonance analyzer, the proton resonance frequency of which was 23.2 MHz, to collect the signal after the sample was kept at 32 ℃ for 10 min. The detection parameters were set as follows: the sampling frequency (SW) = 250 kHz, the radio frequency delay (RFD) = 0.002 ms, the 90° pulse width (P1) = 5 μs, the 180° pulse width (P2) = 10 μs, the sampling points (TD) = 480016, the data radius (DR) = 1, the repeat waiting time (TW) = 3000 ms, the repeated accumulation number (NS) = 4, the echo time (TE) = 0.3 ms, and the echo count (NECH) = 8000.

### 2.9. Statistical Analysis

All experiments were performed in triplicate. The data are shown as mean ± SE. Statistical analyses and analysis of variance (ANOVA) were performed with SPSS 26.0. Statistical significance was regarded at *p* < 0.05. The plots of results were drawn with Origin 19.0.

## 3. Results and Discussion

### 3.1. POV

The POV was an essential parameter of the initial stages of oxidation [27,28]. As shown in Figure 1, the POVs of three oils increased from 0 h to 6 h and decreased from 6 h to 8 h significantly (*p* < 0.05). The POV of 100%DDAG was higher than 0%DDAG and 20%DDAG, indicating that the heating time and DAG content in oil was related to thermal oxidation stability. It is well known that DAG has poor oxidation stability [10,11]. This might explain why the POV increased with the DAG content increment in the hybrid oils. Furthermore, a dynamic balance between peroxides formation and decomposition probably existed in the heating process, and this caused the POV to first increase and then decrease. To be specific, in the early heating period, the oils were easy to be oxidized, and the peroxides formed faster than they dissociated. Thereafter, the reaction balance caused the POV to approach the peak. In contrast to the early heating period, the consumption of the peroxides was more rapidly leading to a decrease in the POV in the late heating phase [29,30]. Chen et al. reported that the POV of palm oil also first increased and then decreased with heating time, which accords with our finding [31].

Furthermore, at the same heating time, the POV under ordinary heating was in most cases higher than HVEF heating (*p* < 0.05), suggesting that HVEF heating could reduce the production of oxides. Tavakoli et al. found that electrostatic field could lead to a 45% drop in POV in soybean oil [23]. The POV of sunflower oil also had a significant decreasing tendency after HVEF treatment [32]. The reason might be that HVEF caused electrons to be transferred to the electrode interface, resulting in a lack of electrons in the oxidation reaction, and finally affecting the generation of oxidative products.

### 3.2. Color

Color, as an important quality parameter of oils, directly affects the perception and acceptability of the consumer [33]. As shown in Table 2, after ordinary heating and HVEF heating of the three duck oils, the *L** decreased significantly (*p* < 0.05), while the *a** and *b** increased significantly (*p* < 0.05) from 0 h to 8 h, suggesting the color of the three duck oils became dark within the heating period. At the same heating time, the *L** of 0%DDAG was higher than that of 20%DDAG and the *L** of 20%DDAG was higher than that of 100%DDAG in most cases, indicating that the color also became turbid with more DAG in the oils.

Furthermore, the *L** of the three oils under HVEF heating was higher (*p* < 0.05) than that under ordinary heating at the same heating time for the most part, especially in the later heating period, implying that HVEF heating could slow down the change of color into darkness. During the heating process, the oils could proceed the thermal oxidation reaction to produce conjugated dienes [34] which could convert the color of oils from yellowness to brownness. A previous study found with the increase in conjugated diene, the brightness gradually decreased in the oil [35]. Adebi et al. reported that pigments of the oils would diffuse toward the surface electrodes in the HVEF [32]. Therefore, our finding might well be explained by the fact that conjugated diene was collected on the electrode surface in the HVEF.

### 3.3. Gas Chromatography

The unsaturation degree of fatty acids is a critical indicator to measure the thermal oxidation stability of oil [36]. As shown in Table 3, the fatty acids composition of the three oils under heating at 180 ℃ showed obvious differences with an extended heating time; the content of saturated fatty acids and trans fatty acids increased, while the content of unsaturated fatty acids decreased with the elongation of heating time. The fatty acid content change tendency of this study was nearly similar to the findings of Casal et al. [37] and Xu et al. [36].

In addition, at the same heating time, the content of saturated fatty acids and trans fatty acids under ordinary heating was higher than that under HVEF heating, suggesting that HEVF heating could reduce the oxidation of unsaturated fatty acids to saturated fatty acids or trans fatty acids. However, Tavakoli et al. reported that HVEF did not change the fatty acid profile in sunflower and soybean oils [23]. This inconsistency might be due to the fact that the heating time and temperature of HVEF were different. In the study of Tavakoli et al., HVEF was used at 65 ℃ for 2 min; the shorter time and the lower temperature might not cause significant changes in fatty acids [25]. It is well known that trans fatty acids have at least one or more double bonds in the trans position [38,39]. Hence, HVEF might affect the formation of trans fatty acids through retarding the production of trans double bonds.

### 3.4. LF-NMR

LF-NMR, as a brand new nondestructive detection technique, plays a significant role in the analysis of oil quality [40]. Figure 2 shows the LF-NMR transverse relaxation time curves (T_2_) of three concentrations of duck oils after HVEF heating and ordinary heating for 8 h. The transverse relaxation time (T_21_) and the peak area ratio (S_21_) calculated from the curves are detailed in Table 4. As seen from Figure 2, the T_21_ showed a progressive shift to the shorter relaxation time with the extension of heating time. In addition, Table 4 shows that the T_21_ of 0%DDAG was longer than that of 20%DDAG, and that of 20%DDAG was longer than that of 100%DDAG. Meanwhile, the S_21_ of three duck oils had an increasing tendency during the heating period, indicating that bigger oxidized polymers or other complicated reaction products were formed and accumulated during the heating process. The T_21_ of the three oils under ordinary heating was lower (*p* < 0.05) than that under HVEF heating, while for S_21_ the situation is just the contrary, suggesting HVEF could exert a negative influence on the generation of oxidation products and total polar compounds.

Various oxidation products and total polar compounds, such as aldehydes, ketones, alcohols, etc., were produced in oils during the high-temperature heating process [29,41,42,43]. When these small molecules accumulated to a certain content, intermolecular interactions were enhanced and the binding force between protons was increased. Therefore, the proton lateral relaxation process took a shorter time to return to the equilibrium state [44].

## 4. Conclusions

During the high-temperature heating process, many peroxides were produced in the oils, which caused *a** and *b**, and the content of saturated fatty acids and trans fatty acids, to increase. Since these lipid peroxides were constantly accumulated, the T_21_ decreased, and S_21_ increased with the lengthening of heating time. The results of POV, color, fatty acid content and LF-NMR show that the oxidation stability was likely negatively correlated with the DAG content in the oils. Moreover, it was indicated that HVEF could effectively slow down the oxidation reaction and delay the process of oil deterioration. Further research could be conducted to investigate the feasibility of HVEF in the frying process.

## Figures and Tables

**Figure 1 foods-11-01322-f001:**
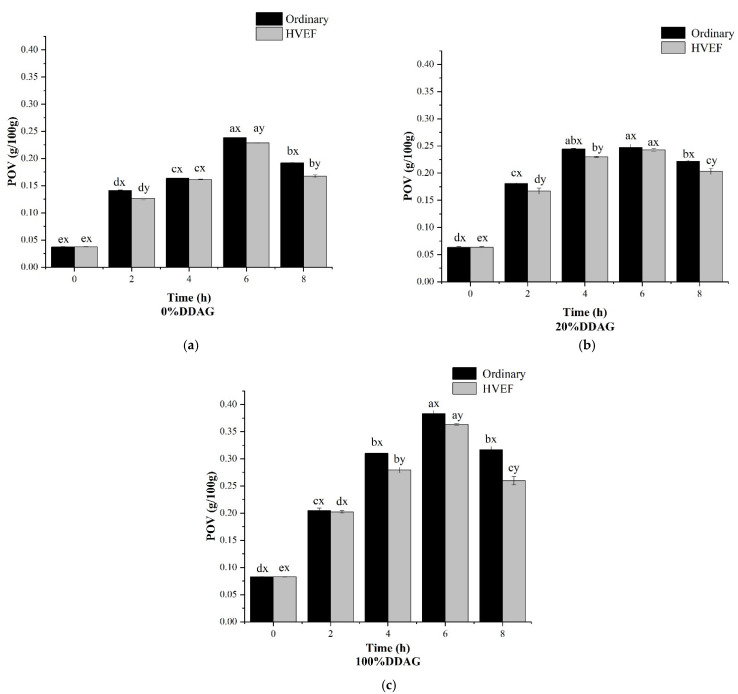
The POV of different DDAG concentrations of duck oils during ordinary heating and HVEF heating at 180 ℃. (**a**) 0%DDAG; (**b**) 20%DDAG; (**c**) 100%DDAG. The letters a–e indicate significant differences (*p* < 0.05) in the same color column at different heating times and the letters x and y indicate significant differences (*p* < 0.05) at the same heating time between ordinary heating and HVEF heating.

**Figure 2 foods-11-01322-f002:**
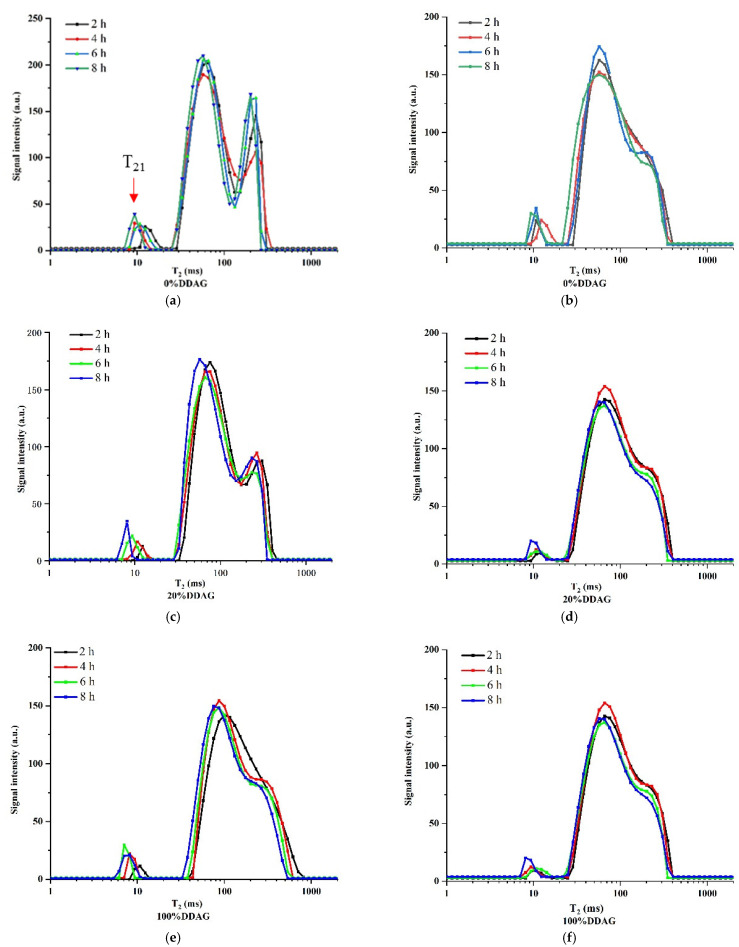
The LF-NMR transverse relaxation time curves of different DDAG concentrations of duck oils during ordinary heating (left) and HVEF heating (right) at 180 ℃. (**a**,**b**) 0%DDAG; (**c**,**d**) 20%DDAG; (**e**,**f**) 100%DDAG.

**Table 1 foods-11-01322-t001:** The information of three hybrid duck oils used in this study.

Oils	DAG Content
0%DDAG	0
20%DDAG	17%
100%DDAG	86%

**Table 2 foods-11-01322-t002:** The color of different DDAG concentrations of duck oils during ordinary heating and HVEF heating at 180 ℃.

Oils	Time (h)	*L**		*a**		*b**	
		Ordinary	HVEF	Ordinary	HVEF	Ordinary	HVEF
0%DDAG	0	81.40 ± 1.34 ^ax^	81.40±1.34 ^ax^	−5.12±1.04 ^ex^	−5.12±1.04 ^ax^	21.54±1.05 ^ex^	21.54±1.05 ^ex^
	2	79.26 ± 0.54 ^ay^	81.37 ± 0.15 ^ax^	−4.04 ± 0.27 ^dx^	−4.42 ± 0.52 ^ax^	28.44 ± 0.32 ^dx^	26.51 ± 0.77 ^dy^
	4	77.38 ± 0.43 ^by^	81.20 ± 0.32 ^abx^	−1.07 ± 0.44 ^cx^	−2.36 ± 0.32 ^by^	50.65 ± 5.84 ^cx^	36.01 ± 0.02 ^cy^
	6	75.53 ± 0.72 ^cy^	79.51 ± 0.18 ^bx^	2.06 ± 0.12 ^bx^	1.05 ± 0.51 ^cy^	65.02 ± 2.66 ^bx^	44.86 ± 2.16 ^dy^
	8	70.89 ± 0.44 ^dy^	78.24 ± 0.32 ^cx^	5.78 ± 0.66 ^ax^	3.93 ± 0.17 ^dy^	73.62 ± 0.66 ^ax^	57.65 ± 1.45 ^ey^
20%DDAG	0	79.29 ± 1.61 ^ax^	79.29 ± 1.61 ^ax^	−5.11 ± 0.37 ^ex^	−5.11 ± 0.37 ^ex^	22.60 ± 0.86 ^ex^	22.60 ± 0.86 ^ex^
	2	73.34 ± 1.10 ^bx^	73.51 ± 1.02 ^bx^	−3.63 ± 0.21 ^dx^	−4.12 ± 0.13 ^dy^	37.27 ± 1.03 ^dx^	33.43 ± 1.21 ^dy^
	4	66.47 ± 0.12 ^cy^	68.99 ± 0.83 ^cx^	1.35 ± 0.58 ^cx^	−0.13 ± 1.35 ^cy^	57.35 ± 1.20 ^cx^	49.51 ± 0.84 ^cy^
	6	62.28 ± 0.49 ^dy^	64.63 ± 0.72 ^dx^	4.39 ± 0.14 ^bx^	3.70 ± 0.65 ^by^	61.75 ± 1.63 ^bx^	55.64 ± 1.19 ^by^
	8	57.06 ± 0.26 ^ey^	60.65 ± 0.79 ^ex^	10.58 ± 0.71 ^ax^	8.59 ± 0.59 ^ay^	77.11 ± 0.12 ^ax^	63.42 ± 0.52 ^ay^
100%DDAG	0	78.27 ± 0.46 ^ax^	78.27 ± 0.46 ^ax^	−2.91 ± 0.69 ^ex^	−2.91 ± 0.69 ^ex^	29.06 ± 1.40 ^ex^	29.06 ± 1.40 ^ex^
	2	71.58 ± 1.10 ^by^	73.40 ± 0.60 ^bx^	−1.11 ± 1.80 ^dx^	−3.54 ± 1.14 ^dy^	49.13 ± 0.88 ^dx^	50.58 ± 1.11 ^dy^
	4	62.17 ± 0.95 ^cy^	66.98 ± 0.39 ^cx^	5.97 ± 1.03 ^cx^	2.56 ± 0.44 ^cy^	62.17 ± 2.51 ^cx^	64.93 ± 0.25 ^cy^
	6	51.57 ± 0.81 ^dy^	59.08 ± 1.89 ^dx^	15.34 ± 0.73 ^bx^	9.30 ± 0.82 ^by^	76.47 ± 1.62 ^bx^	72.46 ± 0.59 ^by^
	8	44.61 ± 1.42 ^ey^	48.46 ± 1.55 ^ex^	18.54 ± 0.49 ^ax^	14.68 ± 0.96 ^ay^	85.08 ± 0.23 ^ax^	76.90 ± 0.12 ^ay^

The letters ^a–e^ indicate significant differences (*p* < 0.05) in the same oil at different heating times under the same heating method, while the letters x and y indicate significant differences (*p* < 0.05) at the same heating time between ordinary heating and HVEF heating. The lightness (*L** value), redness (*a** value) and yellowness (*b** value).

**Table 3 foods-11-01322-t003:** The fatty acid content changes in 0%DDAG, 20%DDAG, and 100%DDAG during HVEF heating and ordinary heating at 180 ℃.

Oils	FA	Ordinary Heating					HVEF Heating				
		0 h	2 h	4 h	6 h	8 h	0 h	2 h	4 h	6 h	8 h
0%DDAG	∑MUFA	44.28 ± 0.05 ^ax^	43.87 ± 0.02 ^by^	43.80 ± 0.03 ^cy^	43.51 ± 0.06 ^dy^	42.58 ± 0.01 ^ey^	44.28 ± 0.05 ^ax^	43.93 ± 0.01 ^bx^	43.84 ± 0.04 ^cx^	43.74 ± 0.03 ^dx^	42.70 ± 0.03 ^ex^
	∑PUFA	21.27 ± 0.09 ^ax^	19.78 ± 0.07 ^by^	18.49 ± 0.12 ^cy^	15.97 ± 0.04 ^dy^	14.66 ± 0.13 ^ey^	21.27 ± 0.09 ^ax^	20.21 ± 0.07 ^bx^	18.75 ± 0.05 ^cx^	16.70 ± 0.03 ^dx^	15.56 ± 0.05 ^ex^
	∑TFA	0.45 ± 0.01 ^ex^	0.66 ± 0.01 ^dx^	0.73 ± 0.03 ^cx^	0.82 ± 0.02 ^bx^	0.98 ± 0.05 ^ax^	0.45 ± 0.01 ^ex^	0.52 ± 0.01 ^dy^	0.63 ± 0.03 ^cy^	0.78 ± 0.05 ^by^	0.85 ± 0.02 ^ay^
	∑SFA	31.26 ± 0.19 ^ex^	32.69 ± 0.23 ^dx^	33.62 ± 0.24 ^cx^	34.05 ± 0.45 ^bx^	35.46 ± 0.11 ^ax^	31.26 ± 0.19 ^ex^	32.32 ± 0.69 ^dy^	33.32 ± 0.25 ^cy^	33.47 ± 0.36 ^by^	34.86 ± 0.13 ^ay^
20%DDAG	∑MUFA	44.90 ± 0.02 ^ax^	44.41 ± 0.03 ^by^	43.93 ± 0.01 ^cy^	43.84 ± 0.04 ^dx^	42.74 ± 0.03 ^ey^	44.90 ± 0.02 ^ax^	44.75 ± 0.01 ^bx^	44.12 ± 0.02 ^cx^	43.88 ± 0.12 ^cx^	43.15 ± 0.02 ^dx^
	∑PUFA	21.58 ± 0.14 ^ax^	19.46 ± 0.07 ^by^	17.77 ± 0.11 ^cy^	15.84 ± 0.06 ^dy^	14.23 ± 0.07 ^ey^	21.58 ± 0.14 ^ax^	19.90 ± 0.05 ^bx^	17.98 ± 0.03 ^cx^	16.31 ± 0.11 ^dx^	14.77 ± 0.09 ^ex^
	∑TFA	0.41 ± 0.01 ^ex^	0.54 ± 0.03 ^dx^	0.65 ± 0.04 ^cx^	0.77 ± 0.03 ^bx^	0.78 ± 0.05 ^ax^	0.41 ± 0.01 ^ex^	0.57 ± 0.04 ^dy^	0.59 ± 0.03 ^cy^	0.61 ± 0.01 ^by^	0.72 ± 0.02 ^ay^
	∑SFA	31.15 ± 0.02 ^ex^	32.07 ± 0.06 ^dx^	33.01 ± 0.11 ^cx^	34.08 ± 0.05 ^bx^	35.14 ± 0.56 ^ax^	31.15 ± 0.02 ^ex^	32.18 ± 0.25 ^dy^	32.80 ± 0.50 ^cy^	33.35 ± 0.16 ^by^	34.13 ± 0.27 ^ay^
100%DDAG	∑MUFA	45.36 ± 0.07 ^ax^	44.92 ± 0.03 ^by^	44.31 ± 0.09 ^cy^	43.96 ± 0.05 ^dy^	43.07 ± 0.12 ^ey^	45.36 ± 0.07 ^ax^	45.21 ± 0.03 ^bx^	44.76 ± 0.06 ^cx^	44.35 ± 0.05 ^dx^	43.68 ± 0.06 ^ex^
	∑PUFA	22.60 ± 0.11 ^ax^	18.77 ± 0.16 ^by^	16.79 ± 0.07 ^cy^	15.57 ± 0.15 ^dy^	13.09 ± 0.09 ^ey^	22.60 ± 0.11 ^ax^	19.82 ± 0.14 ^bx^	16.43 ± 0.26 ^cx^	15.98 ± 0.15 ^dx^	13.91 ± 0.19 ^ex^
	∑TFA	0.35 ± 0.02 ^ex^	0.39 ± 0.03 ^dx^	0.49 ± 0.01 ^dx^	0.55 ± 0.02 ^cx^	0.58 ± 0.03 ^ax^	0.50 ± 0.02 ^ex^	0.58 ± 0.01 ^dy^	0.52 ± 0.02 ^cy^	0.64 ± 0.03 ^by^	0.68 ± 0.03 ^ay^
	∑SFA	29.24 ± 0.38 ^ex^	32.45 ± 0.25 ^dx^	33.34 ± 0.11 ^cx^	35.09 ± 0.12 ^bx^	36.45 ± 0.15 ^ax^	29.24 ± 0.38 ^ex^	31.72 ± 0.33 ^dy^	34.00 ± 0.19 ^cy^	34.02 ± 1.01 ^by^	36.33 ± 0.58 ^ay^

The ∑MUFA means monounsaturated fatty acids; the ∑PUFA means polyunsaturated fatty acids; the ∑TFA means trans fatty acid; the ∑SFA means saturated fatty acids. The letters ^a–e^ indicate significant differences (*p* < 0.05) in the same oil at different heating times under the same heating method, while the letters x and y indicate significant differences (*p* < 0.05) at the same heating time between ordinary heating and HVEF heating.

**Table 4 foods-11-01322-t004:** The T_21_ and S_21_ of 0%DDAG, 20%DDAG, and 100%DDAG during ordinary heating and HVEF heating at 180 ℃ for 8 h.

	Time (h)	0%DDAG		20%DDAG		100%DDAG	
		Ordinary	HVEF	Ordinary	HVEF	Ordinary	HVEF
T21	0	—	—	—	—	—	—
	2	8.92 ± 0.45 ^ay^	9.33 ± 0.00 ^ax^	8.52 ± 0.51 ^ay^	8.92 ± 0.41 ^ax^	7.76 ± 0.45 ^ay^	8.11 ± 0.45 ^ax^
	4	7.76 ± 0.00 ^by^	8.92 ± 0.70 ^ax^	7.40 ± 0.31 ^by^	7.84 ± 0.42 ^bx^	6.75 ± 0.46 ^by^	7.26 ± 0.55 ^bx^
	6	6.75 ± 0.53 ^cy^	7.76 ± 0.61 ^bx^	6.44 ± 0.44 ^cy^	6.75 ± 0.40 ^cx^	5.87 ± 0.41 ^cy^	6.75 ± 0.00 ^cx^
	8	6.44 ± 0.00 ^cy^	7.40 ± 0.41 ^bx^	6.14 ± 0.35 ^cy^	6.53 ± 0.52 ^cx^	5.61 ± 0.51 ^cy^	5.87 ± 0.41 ^dx^
S21	0	—	—	—	—	—	—
	2	2.54 ± 0.15 ^dx^	1.91 ± 0.16 ^dy^	1.74 ± 0.12 ^dx^	0.60 ± 0.15 ^dy^	1.87 ± 0.18 ^dx^	1.57 ± 0.11 ^dy^
	4	3.02 ± 0.16 ^cx^	2.63 ± 0.18 ^cy^	2.49 ± 0.15 ^cx^	1.57 ± 0.12 ^cy^	2.59 ± 0.12 ^cx^	2.23 ± 0.15 ^cy^
	6	4.07 ± 0.21 ^bx^	3.12 ± 0.15 ^by^	3.19 ± 0.16 ^bx^	2.18 ± 0.17 ^by^	3.21 ± 0.15 ^bx^	2.66 ± 0.14 ^by^
	8	4.14 ± 0.15 ^ax^	3.31 ± 0.20 ^ay^	4.15 ± 0.18 ^ax^	3.23 ± 0.15 ^ay^	3.48 ± 0.16 ^ax^	3.01 ± 0.18 ^ay^

The “—” means no T_21_ and S_21_ were detected. The letters a–d indicate significant differences (*p* < 0.05) in the same oil at different heating times under the same heating method, while the letters x and y indicate significant differences (*p* < 0.05) at the same heating time between ordinary heating and HVEF heating.

## Data Availability

The data generated from the study are clearly presented and discussed in the manuscript.

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
