# Peer review of "Effect of High-Voltage Electrostatic Field Heating on the Oxidative Stability of Duck Oils Containing Diacylglycerol"

_foods, 2022, doi:10.3390/foods11091322_

Round 1
Reviewer 1 Report
Dear Authors,
I had an opportunity to review your article. The research is interesting, and the obtained results can be used for further investigation. Nevertheless, the manuscript needs some corrections.
Comments:
Page 3, line 110: what was the concentaration of sodium thiosulfate solution ? it was changed depends of sample heating time?
Page 4, line 146: why no more advanced statistical research was carried out? PCA?
Line 182: please remove comma after Table.
Line 225: the signature T21 is commonly used?
Table 3 and fatty acid component analysis: For the analysis of fatty acid composition after the heating process, it would be more appropriate to use analysis with the addition of an in-house standard. Why was such an analysis not carried out? The percentage share says a little. When the sample is heated, all fatty acids are degraded, cyclic compounds and polymers are formed. The percentage of the sample of fats will not be 100%.
Author Response
Dear editor and reviewer,
Please see the attachment!

Reviewer 2 Report
The manuscript 'Effect of high voltage electrostatic field heating on the oxidative stability of duck oils containing diacylglycerol' meets scientific merit and it is within the scope of the journal. The authors investigated different concentrations of duck oil-based diacylglycerol in terms of peroxide value, color, fatty acids composition and transverse relaxation time curves. The manuscript needs to be revised according to the following comments to achieve its impact and quality.
- Lines 6 -12, 'Affiliation 1; 2; 3; 4; and 5;' should be removed.
- Introduction: Line 58; 'The application of HVEF could help improve the oil refining process [23]'. This information should be expanded with more citations to understand the oil refining process by HVEF.
- The format of the 'Sections' 2.2 through 3.1 should be in accordance with the Journal's Intructions for Authors' The first letters of the words should be Uppercase like 'Section' 3.3.
- Line 124, 'violently' should be rigorously.
- Line 127, 'was used for analyzing' should be 'was used for the analysis'.
- Line 147, 'The data was shown as Mean+/-SE and analyzed by SPSS 26.0. Statistical significance was regarded at p < 0.05. The plots of results were drawn by Origin 19.0.' The sentence should be revised appropriately. What statistical technique (eg One-way ANOVA, etc) was used for the data analysis?
- Line 162, 'In contrast to the early heating period, t In contrast to the early heating period,' should be revised to avoid repetition. what is t?
- Lines 244 and 245 should be revised for clarity.
- Line 250, 'frying filed' what is filed?
- Conclusion: Additional key findings should be provided with clarity.
- Line 270 or Section 5 should be deleted.
- The format of the references list should be in accordance with the Journal's Intructions for Authors'.
- The manuscript should be proofread for clarity of the information presented.
Author Response
Dear editor and reviewer,
Please see the attachment.

Round 2
Reviewer 2 Report
The authors have revised the manuscript according to the comments raised.
However, during the proofreading stage:
- The heading for Section 2.5 should be the same for Section 3.1.
- The journals names should be abbreviated in the list of references.